# Dual Studies of Photo Degradation and Adsorptions of Congo Red in Wastewater on Graphene–Copper Oxide Heterostructures

**DOI:** 10.3390/ma16103721

**Published:** 2023-05-14

**Authors:** Mohamed Rashad, Saloua Helali, Nagih M. Shaalan, Aishah E. Albalawi, Naifa S. Alatawi, Bassam Al-Faqiri, Mohammed M. Al-Belwi, Abdulrhman M. Alsharari

**Affiliations:** 1Physics Department, Faculty of Science, University of Tabuk, Tabuk 71491, Saudi Arabia; 2Physics Department, Faculty of Science, Assiut University, Assiut 71516, Egypt; 3The Center of Energy Research and Technology (CRTEn), Hammam-Lif 2050, Tunisia; 4Department of Physics, College of Science, King Faisal University, P.O. Box 400, Al-Ahsa 31982, Saudi Arabia; 5Department of Biology, Faculty of Science, University of Tabuk, Tabuk 47913, Saudi Arabia

**Keywords:** graphene, copper, Congo red, photocatalytic degradation, adsorption studies

## Abstract

This work comprehensively studies both the photocatalytic degradation and the adsorption process of Congo red dye on the surface of a mixed-phase copper oxide–graphene heterostructure nanocomposite. Laser-induced pristine graphene and graphene doped with different CuO concentrations were used to study these effects. Raman spectra showed a shift in the D and G bands of the graphene due to incorporating copper phases into the laser-induced graphene. The XRD confirmed that the laser beam was able to reduce the CuO phase to Cu_2_O and Cu phases, which were embedded into the graphene. The results elucidate incorporating Cu_2_O molecules and atoms into the graphene lattice. The production of disordered graphene and the mixed phases of oxides and graphene were validated by the Raman spectra. It is noted from the spectra that the D site changed significantly after the addition of doping, which indicates the incorporation of Cu_2_O in the graphene. The impact of the graphene content was examined with 0.5, 1.0, and 2.0 mL of CuO. The findings of the photocatalysis and adsorption studies showed an improvement in the heterojunction of copper oxide and graphene, but a significant improvement was noticed with the addition of graphene with CuO. The outcomes demonstrated the compound’s potential for photocatalytic use in the degradation of Congo red.

## 1. Introduction

There are still outstanding questions that should be prioritized in the relevant academic disciplines concerning the safe drinking and cleaning of water. Recently, it has become an important goal to develop sustainable, supportable nanomaterials with unique benefits such as increased efficiency, selectivity, and practicable stability [1,2,3]. It has been determined that it is now necessary to introduce and create highly effective and sensitive approaches to remove pollutants from water. Thus, it has been established that one of the efficient methods for organic contaminants is catalytic oxidation [4]. Organic and inorganic dyes are used in various industries, including those that deal with leather, paper, plastics, textiles, food processing, printing, cosmetics, and pharmaceuticals. When these colors are found in wastewater, they can also adversely affect other people’s aquatic environments. Both humans and animals who consume these dyes run the risk of developing cancer and other major health issues [5]. It is well-known that using sunshine to photolyze wastewater is an energy-efficient technique, particularly for removing organic dye [6]. Since photocatalysis has numerous uses, including the creation of hydrogen, the elimination of heat-resistant contaminants, and the self-cleaning of surfaces, it is widely known that it has attracted considerable interest in the usage of semiconductor materials. However, metal oxides’ broad bandgap might limit their capacity to absorb visible light. As a result, scientists are working very hard to enhance these materials using substances such as metallic or non-metallic activators [5,6,7]. Graphene is recognized as a good choice for photocatalyst carriers that exhibit excellent performance or promoters, due to its distinctive structure of two-dimensional sheets of carbon atoms, enhanced electron mobility, and localized aromatic system with a high specific surface area, as well as a high chemical and electrochemical stability [8,9]. The surface area of laser-induced graphene was investigated in the literature and exhibited values between 340 and 350 m^2^/g [10,11]. CuO has been studied because it is a strongly correlated electron system [6] and because of its relationship to the Cu-O planes in high-temperature superconductors [12]. Among the 3d transition element monoxides, CuO is distinctive in structure and magnetic properties. In contrast to the cubic shape of other 3d transition metal monoxides, CuO has a low-symmetry monoclinic crystal structure. Particle size effects [13] and uncompensated charges [13,14,15,16], as well as the presence of inherent defects, such as cation or anion vacancies, have all been suggested as explanations for the disparate results in different previous reports [17]. Catalytic properties are influenced by the particles’ size, shape, and crystallinity [18,19,20]. Due to their significant features, including their large surface area, materials composed of carbon buttons, such as graphene and low graphene oxide (RG), are key materials in many domains, including energy storage applications, nanosensors, and lithium-ion batteries [21,22]. Congo red is a chemical with a complex, resilient aromatic composition to degradation. Thus, it is critical to figure out how to remove it from wastewater. This dye is regarded as one of the primary contaminants in the water used by the textile industry [23]. The band gap and UV–Vis absorption of nanostructures make them excellent dye degradation materials. However, there is still work due to the charge carriers’ rapid recombination and limited separation efficiency for electron/hole pairs. Numerous materials have been subjected to fruitless attempts to increase their catalytic activity using dopants and other nanostructures [24,25]. Utilizing materials with heterogeneous interactions was another one of these initiatives, and it significantly improved dye breakdown. Enhancing the photocatalytic activity in dye degradation requires using heterojunction-based substances and activators [26]. It is important to consider the basic workings of photogenerated carrier recombination and how this process affects photocatalytic performance. Therefore, it may be worthwhile to concentrate on the primary methods for controlling the behavior of photogenerated carriers. These photogenerated carriers include heterojunction design, intramolecular donor–acceptor (D–A) system creation, exciton regulation, and electron-spin regulation [27]. The motivation behind the separation and transfer of photogenerated carriers is these photogenerated carriers. Due to their distinctive characteristics, single-site metal atoms or clusters (SMCs) hold a great deal of promise for enabling the investigation of the kinetics and energetics in heterogeneous photocatalysis. In order to investigate the impact of heterojunction and graphene content on the photocatalytic performance of Congo red (CR), CuO–graphene heterostructure nanocomposites are synthesized in this work. The structural and Raman invitations are examined. To clarify the impact of the binary composition, the efficiency is estimated for a variety of compositions. Moreover, the first-order reaction is used for studying the catalytic performance.

## 2. Materials and Methods

### 2.1. Materials Preparation

A solution of CuO was prepared based on the prescribed CuO amount and using deionized water. An amount of 25 mg of CuO was added to 5 mL of deionized water. This concentration corresponds to 63 mM of CuO. The aqueous CuO was sonicated well before withdrawing the required amount. The CuO particles were well-suspended in the solution. Using a CO_2_ laser machine, graphene was fabricated. Firstly, after being cleaned, 8.0 W of power was applied to a polyimide film with a thickness of 170 µm (5.0 mil) at a speed of 100 mm/s, forming a graphene layer of 10 mm × 50 mm. Then 0.5, 1.0, and 2.0 mL of CuO solution were dropped on a graphene sheet (10 mm × 50 mm), which dried on a hot plate at 90 °C. After drying, the sheet was again subjected to the laser beam in the same conditions. The fabrication steps are shown in Figure 1. Based on the experiments, four samples were prepared with pristine graphene (Gr) and CuO-doped graphene of CuO0.5@Gr, CuO1.0@Gr, and CuO2.0@Gr, corresponding to the CuO amount deposited on the graphene sheets.

The absorbance spectrum of CR was measured and exhibited a mean peak at 490 nm. Additionally, there was a small peak at 970 nm [28], as mentioned in the insert of Figure 1. Then the calibration curve between the absorbance and the CR dye concentration was drawn using various dye concentrations, ranging from 0.0 to 30 mg/L [28,29]. The concentration and absorbance were determined to have a straight-line relationship with a correlation coefficient of R^2^ of 0.99, as shown [28]. The dye concentration was directly provided during the UV–Vis spectrum measurement, based on the calibration curve and the spectral absorbance of CR dye as inserted in Figure 1 [30].

Based on the previous calibration curve, the CR concentration was determined after each time. Both photocatalytic degradation and adsorptions of CR on the Gr, CuO0.5@Gr, CuO1.0@Gr, and CuO2.0@Gr samples were studied individually. Photocatalytic degradations of 5 mg/L CR were conducted on 2 mg of Gr, CuO0.5@Gr, CuO1.0@Gr, and CuO2.0@Gr samples with different irradiation times (0–180) min. Using a UV lamp and the adsorption degradation of CR using Gr, CuO0.5@Gr, CuO1.0@Gr, and CuO2.0@Gr samples with different shaking times (0–180) min. were studied.

### 2.2. Characterizations 

Using a monochromatized CuK radiation source (λ = 0.154 18 nm) and an X-ray diffraction (XRD) diffractometer (on a Shimadzu XD-3A, Kyoto, Japan), measurements were formed for the film. A charge-coupled detector and confocal Raman spectroscopy (Lab RAM-HR800, HORIBA, Kyoto, Japan) were also used to examine the products (CCD). The excitation light employed was HeNe, with a wavelength of 633 nm and output power of 20 mW. An arrangement of backscattering at room temperature with a spectral resolution of 0.8 cm^−1^ was employed to measure the Raman spectra. Diffuse reflection occurs when light, other waves, or particles are reflected from a surface so that, unlike in specular reflection, rays incident on the surface are scattered at several angles. When observed from all directions in the half-space next to a diffuse reflecting surface, there is equal brightness, referred to as a Lambertian reflection. A Cary 500 UV–Vis–NIR spectrophotometer was used to measure the diffuse reflection spectra (R) of the samples. Congo red (CR), C_16_H_18_ClN_3_S, was dissolved in 1.0 L of water, and a concentration of CR dye of 5 mg/L was created. 

A set amount of graphene with an oxide variable concentration of 2 mg was used. The solution was moved to a UV lamp, and then the concentration was measured with a time range of 0 to 180 min, as illustrated in Figure 2. The produced solution was added to a model of a shaking instrument (Lab. Tech) for various periods from 0 to 180 min. The bottle was shaken at a constant rotation speed of 150 rpm. A Jenway 6300 UV-Visible spectrophotometer was used to measure the solution’s concentration of CR dye at room temperature at a 490 nm wavelength.

## 3. Results and Discussion

### 3.1. XRD Investigations 

Figure 2 shows the XRD analysis of the prepared films. The XRD spectra were recorded between 2Ɵ = 5 and 70°. A broadening peak between 13 and 30° indicates the graphite peak of the (002) plane [26]. No other peaks were observed for pristine graphene. 

The peaks indexed on the CuO-based materials were very hard to observe for the CuO0.5@Gr sample. However, two peaks at 36.68° and 43.63° were observed for the Cu_2_O phase of the (111) and (200) planes for the CuO1@Gr sample [31,32]. These two peaks were also observed for the CuO2.0@Gr, in addition to a new peak at 50.68° for the Cu(2000) plane [32,33]. The XRD confirmed that the laser beam also was able to reduce the CuO phase to Cu_2_O and Cu phases. This result elucidates incorporating Cu_2_O molecules and atoms into the graphene lattice. The CO_2_ laser was commonly used for transferring graphene Oxide (GO) to reduced GO in different atmospheres [34,35]. The high thermal energy conveyed by the laser selectively etched the irradiated zones, leaving an rGO pattern [36]. The reduced GO surface was also obtained by laser irradiation in both air and N_2_ by Sokolov [37]. Thus, it is expected that the CuO was reduced to Cu_2_O or Cu during the treatment process with the CO_2_ laser.

### 3.2. Raman Spectroscopy 

Figure 3 shows Raman spectra of the pristine graphene and the CuO-doped graphene. Here, we focused on the D-, and G- bands of the Raman spectra. The G expresses the graphite bonds’ presence in the graphene and is associated with the E2g vibrations of the sp^2^ bonds in the Brillouin zone center [38]. In contrast, the D expresses the lattice defects of graphene, which can be observed in the pristine graphene and appears at 1358 cm^−1^ [39]. It is noted from the spectra that the D site changed significantly after the addition of doping, which indicates the incorporation of the CuO in the graphene. Table 1 shows a more detailed analysis of all the Raman spectra parameters, showing the shifts of the three peak positions, and the full-width half-maximum (FWHM) of the peak. The D-bands were observed at 1351.3, 1350.0, 1349.7, and 1344.9 cm^−1^ for the Gr, CuO0.5@Gr, CuO1.0@Gr, and CuO2.0@Gr samples, respectively. The positions of the G-bands were also observed at 1582, 1580.5, 1580.4, and 1580.3 cm^−1^ for the Gr, CuO0.5@Gr, CuO1.0@Gr, and CuO2.0@Gr samples, respectively. The positions of all peaks were changed after adding CuO. The FWHM of the pristine Gr and the CuO-doped graphene peaks are shown in Table 1. The FWHM increased upon adding the CuO, indicating the incorporation of Cu molecules into the graphene. The FWHM indicates a disorder in the chains of the graphene lattice. The intensity ratios (I_D/G_) of the D and G peaks were calculated to be 0.31, 0.33, 0.40, and 0.84 for the pristine graphene and the CuO-doped graphene in the sequences. These results indicate that with increasing the CuO content, more defects appear in the graphene lattice [40,41,42]. The obtained results of the Raman spectra present a synergetic effect of CuO on graphene when prepared using the CO_2_ laser method. The intensity ratios (I_2D/G_) of the 2D and G peaks were calculated and found to be less than 1.0, confirming the formation of multilayer graphene [43]. The value of I_2D/G_ decreased from 0.47 to 0.19 when adding CuO.

### 3.3. Optical Investigations 

The spectra of the diffuse-reflectance (R) versus wavelength (λ) for the pristine Gr and the CuO-doped graphene (Gr, CuO0.5@Gr, CuO1.0@Gr, and CuO2.0@Gr samples) are shown in Figure 4. At wavelengths between 200 and 300 nm, the R values were small for almost all samples in the first region. Then they rose exponentially with the increasing wavelength in the second region (260–350 nm). Finally, they stabilized for the further wavelength increase in the third region (>350 nm). Additionally, the value of R increased as the percentage of CuO increased from 0.5 to 2% in the graphene. The reduction of graphene oxide is expected to increase the intensity of the C–C bonds since the annealing of the structure happens through the reduction of carbon–heteroatom bonds [44]. The following equation gives the value of *F*(*R*) using the Kubelka–Munk theory [45]:(1)$$F\left(R\right)=\frac{{\left(1-R\right)}^{2}}{2R}$$

Figure 5 displays the *F*(*R*) for the examined Gr, CuO0.5@Gr, CuO1.0@Gr, and CuO2.0@Gr samples. The *F*(*R*) value for all investigated samples decreased as the percentage of CuO was raised over the analyzed wavelengths.

Figure 6 shows a plot of (F(R)hυ)^2^ versus the photon energy (hυ) for Gr, CuO0.5@Gr, CuO1.0@Gr, and CuO2.0@Gr samples to obtain the optical band. The electronic calculations of the graphene revealed that the band gap was 0.952 eV, proving that graphene monoxide is a semiconductor with a direct band gap [46]. Our results reveal that the fitting line of these curves obtains the optical band gaps, which are 0.4, 0.73, 1.06, and 0.90 eV, for the Gr, CuO0.5@Gr, CuO1.0@Gr, and CuO2.0@Gr samples. 

### 3.4. Catalyst Studies 

After the previous studies of the Gr, CuO0.5@Gr, CuO1.0@Gr, and CuO2.0@Gr samples, we will modify both the photocatalytic efficiency and the adsorptions of Congo red (CR) dye using these samples. 

#### 3.4.1. Photocatalytic Studies 

The efficiency of CR on the Gr, CuO0.5@Gr, CuO1.0@Gr, and CuO2.0@Gr samples was calculated using the following equation [47]: (2)$$\eta \%=\frac{C_{o}-C_{t}}{C_{o}}\times 100$$
where *C_o_* is the initial concentration of CR (mg L^−1^) and *C_t_* is the concentration of CR (mg L^−1^) at time *t*. The photodegradation efficiencies of CR through the Gr, CuO0.5@Gr, CuO1.0@Gr, and CuO2.0@Gr samples in 180 min in UV light were 18.66%, 32%, 21.33%, and 26.33%, respectively, as shown in Figure 7. CuO2@Gr shows the highest efficiency in the photodegradation study of CR dye.

The photodegradation rate constants of CR dye were determined by the kinetics model of Langmuir–Hinshelwood [48] using the following equation:(3)$$\ln\left[\frac{C_{t}}{c_{0}}\right]=-kKt+KC_{0}$$
where *k* is the first-order reaction rate constant and *K* is the adsorption equilibrium constant.

The first person to explain how a bimolecular reaction occurs at a catalyst’s surface was Langmuir [49,50]. He recognized two types of surface interactions: (i) the Langmuir–Hinshelwood (L–H) mechanism, which is an interaction between molecules or atoms adsorbed at adjacent sites on the surface; and (ii) the Eley–Rideal (E–R) mechanism, which is an interaction that results from the collision of the gas molecules of one of the reactants and the adsorbed molecules of the other. The Langmuir–Hinshelwood mechanism and pseudo-first-order kinetics are used to describe the degradation, as shown by the linear and logarithmic plots in Figure 8.

Table 2 summarizes the determined values of the degradation equilibrium constant (*K*) and the degradation rate constant (*k*) for various composites. The results show that both the *k* and *K* constants depend on the weight percentage of the CuO, where *k* varies from 0.70549 down to 0.31863 and *K* varies from 0.001517 up to 0.00353 for Gr, CuO0.5@Gr, CuO1.0@Gr, and CuO2.0@Gr, respectively. Remarkably, the CuO0.5@Gr shows the best R-square value, indicating the highest photodegradation efficiency value.

#### 3.4.2. Photocatalysis Mechanism

As we mentioned before, two phases are present in our samples; therefore, in the case of the CuO phase, based on the above observations, the proposed photocatalyst is more effective for CR degradation owing to its lower concentration of CuO. Moreover, a higher weight percentage of CuO (>0.5 mol %) decreases photodegradation efficiency. Copper ions (Cu^2+^/Cu^+^) and (Cu^2+^/Cu) with redox potentials of 0.16 V and 0.52 V, respectively, versus the normal hydrogen electrode (NHE), can trap photogenerated electrons [47]. Therefore, the degradation could happen because samples with higher CuO concentrations have a “downward shift” in the relative location of the conduction band bottom. The reduction potentials vs. the SHEs of Cu^2+^ (0.52 V) and Cu^+^ (0.34) are greater than those of graphene (+0.22 V). Therefore, Cu^2+^ will first react with graphene and then cause electron transfer [46]. Essentially, the electron is transferred from the graphene into the Cu. The electron produced from the excitation of a photocatalyst is immediately captured by the Cu^2+^, which frees up the photocatalyst’s oxidative valance hole to oxidize organic molecules [51]. This mechanism is based on recognizing that graphene functions as a catalyst and electron transport medium. On the other hand, in the case of the Cu_2_O phase, the mechanism of the photodegradation of CR on the surface of CuO@Gr may be expected in terms of the partial oxidation of graphene and the Cu_2_O phase. Based on the optical properties of the current samples and available data in the literature [52,53,54], the band structure is proposed in Figure 9a. Cuprous oxide (Cu_2_O) is a p-type semiconductor with low electron affinity and very high hole mobility that has potential for hole transport. Hence, it is expected that the partial oxidation may have caused the narrow band of 0.4 eV that was observed in the optical measurements of the pristine graphene. With the incorporation of Cu_2_O in the graphene lattice, this band gap was extended to reach 1.0 eV in a different position in the graphene. Thus, the CR photodegradation can be explained by the sight of this band structure. When the Cu_2_O@Gr composite is photoexcited, an ${(e}^{-}-h^{+})$ pair is formed, and then the photoreaction between the dye and water species begins [46,54,55,56]. According to the band structure of Figure 9b, the photoexcited electrons are allowed to transfer to the graphene conduction band, which may increase the photocatalysis performance of the doped graphene. Thus, a direct oxidation reaction occurs on the surface due to the oxidizing potential of the $h^{+}$:(4)$$h^{+}+dye\to {dye}^{.+}\to oxidation\;of\;the\;dye$$

The hydroxyl radicals (*OH·*) are then formed by the decomposition of water molecules and the *OH*^−^ reaction with the exited holes. The *OH*· radicals are responsible for photocatalysis:(5)$$h^{+}+H_{2}O\to H^{+}+·OH$$
(6)$$h^{+}+{OH}^{-}\to ·OH$$

The peroxide anions are also united to form ·OH radicals when peroxide anions are formed by reacting the oxygen molecules with the exited electrons:(7)$$e^{-}+O_{2}\to ·O_{2}^{-}$$

The ·OH radicals are high oxidizers with 2.8 V in potential. Thus, based on the above formation of ·OH radicals, the organic dye degrades. Obtaining high photocatalytic efficiency with a low e-h recombination rate is one of the most important factors for photodegradation. Reducing this rate reduces wasted light energy and increases the yield of ·OH radicals. Thus, increasing the yield increases CR decomposition in water. This required efficiency or higher yield can be achieved through the transfer of photoexcited electrons from the graphene spots doped with Cu_2_O to the other parts, due to the high conduction band of the doped spots. Therefore, the addition of Cu_2_O in the current study increased the band gap of the doped graphene, which prevented e-h recombination. Therefore, the concentration of ·OH was increased in the aqueous solution and led to rapid degradation of the dye.

#### 3.4.3. Adsorption Studies

On the other hand, the adsorption efficiency of CR through the Gr, CuO0.5@Gr, CuO1.0@Gr, and CuO2.0@Gr samples at 180 min shaking time was obtained at 21%, 39%, 55%, and 43.5%, respectively, as represented in Figure 10. Comparing photocatalytic and adsorption processes after 180 min. shows that the adsorption efficiency was higher than the photolytic. Moreover, it confirms that the CuO0.5@Gr shows high performance towards the photodegradation of CR dye.

From more detailed investigations of the above results, the degraded dye capacity (*q_t_* in mg/g) can be calculated using the following equation [44]:(8)$$q_{t}=\frac{\left(C_{0}-C_{t}\right)V}{m}$$
where *V* is the volume of the solution in L and *m* is the mass of dry adsorbent used in mg. Figure 11 illustrates the degradation dye capacity versus the shaking time for concentrations of 5 mg/L of CR. With longer shaking times, the CR dye had more significant potential for showing degraded capacity. The CuO2@Gr sample displayed the weakest surface reactivity to the dye. However, a significant improvement was observed for the other samples. Three models were employed to analyze the kinetics of CR dye adsorption on the Gr, CuO0.5@Gr, CuO1.0@Gr, and CuO2.0@Gr samples: pseudo-first-order, pseudo-second-order, and intraparticle diffusion models.

The following equations can be used to explain these models [57,58]:(9)$$\log\left(q_{e}-q_{t}\right)=logq_{e}-K_{1}\frac{t}{2.303}\;\text{Pseudo-First-Order}\mathrm{Model}$$
where *q_t_* is the quantity of degraded Congo red capacity at time (*t*) measured in mg/g, *q_e_* is the quantity of degraded Congo red at equilibrium state in mg/g, and *K*_1_ is the pseudo-first-order constant in min^−1^.
(10)$$\frac{t}{q_{t}}=\frac{1}{K_{2}q_{e}^{2}}+\frac{t}{q_{e}}\;\text{Pseudo-Second-Order}\mathrm{Model}$$
(11)$$q_{t}=K_{diff}\sqrt{t}+C\;\mathrm{Intraparticle}\mathrm{Diffusion}$$
where *K*_2_ is the pseudo-second-order constant in g/mg·min, *K_diff_* is the intraparticle diffusion kinetic model in mg/g·min^1/2^, and *C* is a constant. Figure 12 illustrates a plot of the three models versus shaking times for Gr, CuO0.5@Gr, CuO1.0@Gr, and CuO2.0@Gr. The slope and intercept of the fitted lines were used to calculate the parameters of the three kinetic models and are listed in Table 3. The values of the correlation coefficients (R^2^) for the pseudo-first-order model for CuO2.0@Gr (R^2^ = 0.97) were found to be more fitted, which can predict the adsorption capacity compared to the corresponding experimental capacity of the present sample. The kinetic intraparticle diffusion model has a high correlation coefficient, indicating that the adsorption of CR on graphene with different concentrations of CuO follows the intraparticle diffusion kinetic model. Moreover, the fact that the diffusion curves did not traverse the coordinate system’s origin shows that surface and intraparticle diffusion played a role in the adsorption mechanism. The Elovic equation is another recommended equation for analyzing the adsorption of the investigated sample, which is expressed as follows [59]:(12)$$q_{t}=\frac{\ln\left(\alpha \beta \right)+ln(t)}{\beta }$$
where *α* is the initial sorption rate in mg/g·min and *β* is the desorption constant in g/mg. The above equation is frequently used in studies of the chemisorption of gases onto solids. Similarly, it can be used to study the solutes that are adsorbing from a liquid solution. The relationship of qt with *ln*(*t*) is represented in Figure 13a. The sorption (*α*) and desorption (*β*) parameters were determined and mentioned in Table 3. Furthermore, it exhibits a linear behavior, which agrees with Elovic’s kinetic model. Generally, the adsorption process can be divided into several processes involving (i) solution bulk transport, (ii) film diffusion, (iii) particle diffusion, and (iv) particle and solid-surface sorption and desorption.

Processes (ii) and (iii) are known as rate-limiting processes when they occur quickly [60]. The adsorption process’s diffusion mechanism was calculated using the following formula [61]:(13)$$B_{t}=-0.4977-ln(1-\frac{q_{t}}{q_{e}})$$

Figure 13b represents the relationship of *B_t_* versus shaking time. The curve shows that straight lines do not pass through the origin point. Hence, diffusion could be considered as a process’s rate-limiting stage.

#### 3.4.4. Dual Studies of Photo Degradation and Adsorptions

In our dual study, the set of Gr, CuO0.5@Gr, CuO1.0@Gr, and CuO2.0@Gr samples was used in the photodegradation and then in the adsorption process of CR in wastewater. Figure 14 shows two individual experiments carried out in two steps for the same set of mentioned samples. (i) The photocatalysis properties were carried out with different irradiation times of up to 180 min, as shown in left part of Figure 14. The same set of the present samples was used for adsorption process with different shaking times of up to 180 min, as shown in the right part of Figure 14. One can see that CuO0.5@Gr reached an excellent efficiency of 75% after 180 min of adsorption studies. The proposed photocatalyst is more effective in CR degradation owing to its lower concentration of copper. This can be attributed to the strong interaction between copper and graphene, where the incorporation of CuO increased the band gap, which may have caused more absorption of incident photons. The photoexcited Cu_2_O could also transfer the excited electron to the graphene, which increased the hydroxyl radicals in the solution. After the irradiation process, the adsorption process took place. The doped samples showed a high adsorption of CR compared to the pristine one. It is expected that the large-sized copper atoms intercalated between the graphene layers, causing a greater adsorption of CR in these sites compared to the sacked graphene layers. This result is promising for future work on the mentioned photo-adsorption cycle processes, where the samples are reusable for photocatalysis and adsorption properties.

## 4. Conclusions

In conclusion, a copper oxide–graphene heterostructure nanocomposite phase was created to examine the impact of the oxide–graphene heterojunction on photocatalytic degradation and adsorption studies of Congo red dye. A CO_2_ laser was used to create the graphene, and CuO was incorporated into the graphene. The effect of the CuO content in the graphene was investigated at 0.5%, 1%, and 2% wt.%. According to the photocatalysis and adsorption measurements, a graphene heterojunction with a suitable amount of CuO led to an improvement. Promising dual studies of both the photodegradation and the adsorption of CR in wastewater on graphene–CuO heterostructures were applied to reach an excellent efficiency of 75%. The findings suggest that this oxide–graphene heterostructure has promise for photocatalytic use in the photodecomposition of Congo red.

## Data Availability

All relevant data are within the paper. The data presented in this study are available on request from the corresponding authors.

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
