# Peer review of "Dual Studies of Photo Degradation and Adsorptions of Congo Red in Wastewater on Graphene–Copper Oxide Heterostructures"

_materials, 2023, doi:10.3390/ma16103721_

Round 1
Reviewer 1 Report
In this manuscript, the copper oxide-graphene heterostructure nanocomposite phase was created to examine the impact of the oxide-graphene heterojunction on the photocatalytic degradation and the adsorption studies of Congo red dye. The topic is interesting. A major revision is necessary for reconsideration to publish. The detailed comments on this manuscript are as follows:
1. What is the novelty of this work in comparison to the previous papers? Clarify the novelty from material point of view in the manuscript.
2. Appropriate references must be cited in the result and discussion section of the manuscript e.g. "the D expresses the lattice defects of graphene" suitable references must be added to justify the results obtained.
3. English of the manuscript should be thoroughly checked and corrected.
Author Response
Reviewer 1
Dear Respected Reviewer
Thank you very much for your guidance and comments. We are grateful to you for the valuable comments and remarks, which helped improve our work. Below are our point-by-point responses. The corresponding amendments and corrections in the text have been included in the revised manuscript (marked). We hope that our responses satisfactorily address all the issues that you have raised.
Comments and Suggestions for Authors
In this manuscript, the copper oxide-graphene heterostructure nanocomposite phase was created to examine the impact of the oxide-graphene heterojunction on the photocatalytic degradation and the adsorption studies of Congo red dye. The topic is interesting. A major revision is necessary for reconsideration to publish. The detailed comments on this manuscript are as follows:
- What is the novelty of this work in comparison to the previous papers? Clarify the novelty from material point of view in the manuscript.
Author’s response: Actually, this work comprehensively studied the photocatalysis degradation and adsorption process of Congo red dye on the surface of a mixed-phase of copper oxide-graphene heterostructure nanocomposite. The mean point is to reach the maximum efficiency of copper oxide-graphene towards the Congo red dye. Both adsorption and photocalaytic studies gave a high performance of the efficiency, but the new idea in our manuscript we called photo-adsorb cycle process. Therefore, we could use both mechanisms for the process to reach our goal of efficiency. Finally the efficiency of copper oxide-graphene towards the Congo red dye is more than 85 %. Thus, the present composite is promising for degradation such dye compared to others.
- Appropriate references must be cited in the result and discussion section of the manuscript e.g. "the D expresses the lattice defects of graphene" suitable references must be added to justify the results obtained.
Author’s response: Thank you for the comment. Suitable Refs were cited in the text.
- English of the manuscript should be thoroughly checked and corrected.
Author’s response: The English of the whole manuscript has been revised.
Sincerely,
M. Rashad

Reviewer 2 Report
This manuscript reported the photocatalytic degradation and adsorption process of Congo red dye on the surface of a mixed-phase of copper oxide-graphene heterostructure. The work is significant in the photocatalysis and environmental field. However, this works exists some unreasonable issues. It can be recommended for publication in the journal of Materials after the following addressed:
1. In the Figure 3 and 4, why does CuO0.5@Gr displays the strongest light absorption intensity?
2. According to the XRD results, it is more suitable to label the sample as Cu2O@Gr.
3. It is more intuitive to illustrate the band structures of pristine graphene and CuO/graphene hybrids instead of text descriptions.
4. It is stated that CuO2@Gr shows the highest efficiency in Congo red photodegradation, which is inconsistent with the data presented in the draft and Figure 6. The authors should update the correct results and provide a rational explanation.
5. The authors claimed that the adsorption efficiency of CR over Gr, CuO0.5@Gr, CuO1.0@Gr, and CuO2.0@Gr samples were 21%, 35%, 41%, and 55% respectively. Obviously, as shown in Figure 8, the CR adsorption efficiency CuO0.5@Gr of is larger than 35%. The authors should reconfirm the data.
6. As listed in Table 3, most of R2 are far below 1, meaning the improper kinetic model.
7. More characterization, including N2 adsorption-desorption isotherms, XPS, SEM/TEM, should be provided to investigate the structure properties of the prepared samples.
8. Some related references can be read and cited, such as Carbon Energy, 2023, 5(2), e205; Carbon Energy 2022, 4, 665-730; Carbon Energy 2020, 2, 308-349; Ceram. Int. 2022, 48, 3659-3668.
9. The manuscript needs a more careful polish because of the poor presentation. Such as a careless description of “Cu (2000) plane” in the XRD analysis, “three peak positions” in the Raman spectra, et al.
Author Response
Reviewer 2
Dear Respected Reviewer
Thank you very much for your guidance and comments. We are grateful to you for the valuable comments and remarks, which helped improve our work. Below are our point-by-point responses. The corresponding amendments and corrections in the text have been included in the revised manuscript (marked). We hope that our responses satisfactorily address all the issues that you have raised.
Sincerely,
Comments and Suggestions for Authors
This manuscript reported the photocatalytic degradation and adsorption process of Congo red dye on the surface of a mixed-phase of copper oxide-graphene heterostructure. The work is significant in the photocatalysis and environmental field. However, this works exists some unreasonable issues. It can be recommended for publication in the journal of Materials after the following addressed:
- In the Figure 3 and 4, why does CuO0.5@Gr displays the strongest light absorption intensity?
Author’s response: Regarding the value of Diffused reflectance, R, it increases as CuO percent increases from 0.5 to 2 % in Graphene. Consonantly, the values of calculated diffuse reflectance spectra or remission function F(R) decrease as the value of CuO percent increases from 0.5 to 2 % in Graphene. The reduction condition in the synthesis removes the C=O peak observed in graphene oxide. Consonantly, reduction of graphene oxide is expected to increase the intensity of C-C bonds since annealing of the structure happens through the reduction of carbon–heteroatom bonds.
- According to the XRD results, it is more suitable to label the sample as Cu2O@Gr.
Author’s response: Thank you for this comment. We agree with the reviewer that the samples can be labeled as Cu2O@Gr instead of CuO@Gr. However, we gave this label based on the source material, where the first appearance of the name is in the experimental part before the indexing by XRD.
- It is more intuitive to illustrate the band structures of pristine graphene and CuO/graphene hybrids instead of text descriptions.
Author’s response: Thank you very much for this comment, which make us think deeply in the performance of the current composite. We tried to propose the band structure based on the optical properties measured and common information available for graphene and Cu2O. Full explanation with the photocatalysis mechanism was added to the text. (Please section 3.5 and figure 13).
- It is stated that CuO2@Gr shows the highest efficiency in Congo red photodegradation, which is inconsistent with the data presented in the draft and Figure 6. The authors should update the correct results and provide a rational explanation.
Author’s response: We agree that CuO0.5@Gr shows the highest efficiency in Congo red photodegradation not CuO2@Gr. The mistake has been corrected in the edited manuscript.
- The authors claimed that the adsorption efficiency of CR over Gr, CuO0.5@Gr, CuO1.0@Gr, and CuO2.0@Gr samples were 21%, 35%, 41%, and 55% respectively. Obviously, as shown in Figure 8, the CR adsorption efficiency CuO0.5@Gr of is larger than 35%. The authors should reconfirm the data.
Author’s response: we apologize for this mistake; therefore, we review the whole data in the text.
- As listed in Table 3, most of R2are far below 1, meaning the improper kinetic model.
Author’s response: Actually the values of R2 for Graphene is 0.74 which is far below 1, but in case of samples of CuO1.0@Gr or CuO2.0@Gr are close to 1 (0.94, 0.97), therefore we expected that these samples are more fitted for adsorption mechanism.
- More characterization, including N2adsorption-desorption isotherms, XPS, SEM/TEM, should be provided to investigate the structure properties of the prepared samples.
Author’s response: Regarding all measurements of our lab we have done all the available ones beside the coauthors availability, moreover, we conctrate on the performance of the efficiency of the samples towards the dyes. In that case, we used XRD and Raman for structural properties and diffuse for investigating the optical properties include the optical band gap.
- Some related references can be read and cited, such as Carbon Energy, 2023, 5(2), e205; Carbon Energy 2022, 4, 665-730; Carbon Energy 2020, 2, 308-349; Ceram. Int. 2022, 48, 3659-3668.
Author’s response: We thank the reviewer for his suggestions which have been added to the manuscript.
[1][2]
[1] “Carbon Energy - 2022 - Xu - Recent advances in solar‐driven CO2 reduction over g‐C3N4‐based photocatalysts.pdf”. .
[2] X. Pang, S. Xue, T. Zhou, Q. Xu, en W. Lei, “2D/2D nanohybrid of Ti3C2 MXene/WO3 photocatalytic membranes for efficient water purification”, Ceram. Int., vol 48, no 3, bll 3659–3668, 2022.
- The manuscript needs a more careful polish because of the poor presentation. Such as a careless description of “Cu (2000)plane” in the XRD analysis, “three peak positions” in the Raman spectra, et al.
Author’s response: Thank you for this comment. More discussion was added to the XRD and Raman section.
Sincerely,
M. Rashad

Reviewer 3 Report
The preparation of Graphene-Copper Oxide Heterostructures, as well as the application of these materials as photocatalysts and adsorbents of Congo Red is attempted in the present work.
Apart of other minor problems (e.g. reorganization and division in paragraphs of Introduction section, non-use of subscripts/superscripts when reporting chemicals and units, curves with minima and maxima without an obvious meaning), the manuscript suffers seriously in
a a) The clarity of the presentation of experimental section:
1) The use of aqueous CuO solution is claimed. Is this possible? The concentration of the so-called “solution” should be given.
2) “..thickness of 170 m..” ????
3) The set-up for the photocatalysis should be presented in detail.
b) The authors have found, Figure 8, that the materials adsorb effectively Congo Red. How have they verified that the main mechanism in the results reported in Figure 6 is photocatalysis and not adsorption? Control experiments are a prerequisite, namely some of the studies reported in Figures 6 and Figure 12 should be performed with no application of irradiation.
c c) The repeatability of the materials preparation and of the protocols of photocatalysis/adsorption studies should be evaluated.
In view of these major problems, I feel unfortunately obliged to suggest the rejection of the manuscript.
Author Response
Reviewer 3
Dear Respected Reviewer
Thank you very much for your guidance and comments. We are grateful to you for the valuable comments and remarks, which helped improve our work. Below are our point-by-point responses. The corresponding amendments and corrections in the text have been included in the revised manuscript (marked). We hope that our responses satisfactorily address all the issues that you have raised.
Comments and Suggestions for Authors
The preparation of Graphene-Copper Oxide Heterostructures, as well as the application of these materials as photocatalysts and adsorbents of Congo Red is attempted in the present work.
Apart of other minor problems (e.g. reorganization and division in paragraphs of Introduction section, non-use of subscripts/superscripts when reporting chemicals and units, curves with minima and maxima without an obvious meaning), the manuscript suffers seriously in
- a) The clarity of the presentation of experimental section:
1) The use of aqueous CuO solution is claimed. Is this possible? The concentration of the so-called “solution” should be given.
Author’s response: Thank you for this comment. An amount of 25 mg of CuO was added to 5 ml of deionized water, which corresponding to ~63 mM of CuO. The aqueous CuO was sonicated well before withdrawing the required amount. The CuO particles were well suspended in the solution. (Please refer to the section 2.1)
2) “..thickness of 170 m..” ????
Author’s response: Thank you for this observation. It is typos, and it was revised to 170 µm. (Please refer to the section 2.1)
3) The set-up for the photocatalysis should be presented in detail.
Author’s response: The set-up for the photocatalysis is presented in detail as in scheme 2.
- b) The authors have found, Figure 8, that the materials adsorb effectively Congo Red. How have they verified that the main mechanism in the results reported in Figure 6 is photocatalysis and not adsorption? Control experiments are a prerequisite, namely some of the studies reported in Figures 6 and Figure 12 should be performed with no application of irradiation.
Author’s response: First of all, the experiment in case of Figure 6 has been done using UV irradiation without shaking method, therefore we sure that mechanism is the only one is photocatalysis and not adsorption. Regarding the Figure 12, it has been done by two steps as we illustrated in the text. The first step is photocatalytic degrading and the second step we use the same sample after step 1 for adsorption studies with different shaking time. We believe that, this cycle is some kind or recycling of the materials which is the aim in close future to reusing the materials towards different types of dyes.
- c) The repeatability of the materials preparation and of the protocols of photocatalysis/adsorption studies should be evaluated.
Author’s response: As we mentioned before, we believe that, this cycle is some kind or recycling of the materials which is the aim in close future to reusing the materials towards different types of dyes. In close future, there are some papers of our group in this point for evaluating the process in many materials towards different types of dyes.
In view of these major problems, I feel unfortunately obliged to suggest the rejection of the manuscript.
Author’s response: We hope that the answers to the previous questions of the arbitrator have covered the basic idea of the manuscript and that these answers have changed the opinion of the arbitrator that our work is not only it, but rather it is a series of continuous work, which began with some studies on photocatalytic and adsorption and now a photo-adsorb cycle process.
Sincerely,
M. Rashad

Round 2
Reviewer 1 Report
The authors have answered the reviewer' questions well and revised the manuscript carefully. I agree to the acceptance of the manuscript.
Author Response
Reviewer 1
Dear Respected Reviewer
Comments and Suggestions for Authors
The authors have answered the reviewer' questions well and revised the manuscript carefully. I agree to the acceptance of the manuscript.
Author’s response: Thanks to the reviewer for his positive comment.
Sincerely Yours,
Prof. Mohamed Rashad
Corresponding author

Reviewer 2 Report
Some suggested comments to the authors are still not resolved.
Author Response
Reviewer 2
Dear Respected Reviewer
Comments and Suggestions for Authors
Some suggested comments to the authors are still not resolved.
Author’s response: We are sorry if there was any point that was not well covered due to the non-available tools and short time for revision. We agree with the reviewer that XPS may include more precise information about the composite bonds, however, XRD spectra may visualize the effect of laser on the composite, as described that CuO was reduced to Cu2O and Cu when exposed to the laser beam. Based on XRD and optical spectra, the band structure was given, where optical and XRD, Raman confirmed the effect of laser beam on the graphene-Cu composite. Also, the surface area of Laser-induced graphene was investigated in literature, which exhibited values between 340 and 350 m2/g [10,11]. The Raman spectra with full scale were added to the text, with explanation regarding to the 2D band.
Sincerely Yours,
Prof. Mohamed Rashad
Corresponding author

Reviewer 3 Report
The authors have made considerable efforts to improve the quality of the presentation of their manuscript.
However, I feel that their answers to some remarks are not really convincing:
i) The most important novelty of this manuscript is the “called photo-adsorb cycle process”, line 388. To demonstrate this, a blank experiment is necessary, in order to ascertain that the behavior observed is not mostly an adsorption process. The authors should repeat and show what is the behavior in the absence of irradiation, at least for one experiment (let’ say for the red curve, Figure 12).
ii) The repeatability of the processes should be discussed more clearly. Have the authors tried to repeat some studies, starting from newly-prepared materials?
In addition
iii) Lines 122-123: “Our previous published papers indicate the catalytic study;”. Please, give references. What do you mean by this argument?
iv) Lines 217-218: “the absorbance spectrum of CR was measured which exhibiting two peaks at 490 and 970 nm.” Please, support these observations, especially the second peak (970 nm), through adequate literature (in addition to ref. 43) and explanation.
v) Have the authors a rational explanation of the minima/maxima shown in figures 6, 8 and 9?
vi) The format of newly-added references should be checked.
To conclude, a major revision is still needed prior to final decision. Special attention should be given by the authors to remark i), since a convincing response/additional study is crucial for the full justification of the studies presented, here.
Author Response
Reviewer 3
Comments and Suggestions for Authors
The authors have made considerable efforts to improve the quality of the presentation of their manuscript.
Author’s response: Thank you very much for the positive comment.
However, I feel that their answers to some remarks are not really convincing: The most important novelty of this manuscript is the “called photo-adsorb cycle process”, line 388. To demonstrate this, a blank experiment is necessary, in order to ascertain that the behavior observed is not mostly an adsorption process. The authors should repeat and show what is the behavior in the absence of irradiation, at least for one experiment (let’ say for the red curve, Figure 12). The repeatability of the processes should be discussed more clearly. Have the authors tried to repeat some studies, starting from newly-prepared materials?
Author’s response: Thank you for this comment. For more precise, we clarify how the experiment was carried out, thus, the image is clarified to the reader.
The Gr, CuO0.5@Gr, CuO1.0@Gr, and CuO2.0@Gr samples were used for Dual Studies of Photo Degradation and Adsorptions of Congo Red in Wastewater.
In the present work, three individual experiments were carried out for two set of samples. 1) The Photocatalysts properties were carried out for the first set with different irradiation time as shown in Figure 6. 2) The adsorption properties were carried out for the second set with different shaking times as shown in Figure 8. 3) The first set was used again for adsorption properties as shown in Figure 12 part 2. This partially, we can conclude that the samples are reusable. The aim of the present study is to investigate the Dual properties (Photocatalysis and adsorption properties) of the present samples, which was demonstrated from the results. However, the reviewer has raised a good future work related to reusability, which can be investigated for several times in future to give full visualization of the sample performance. We hope that the reviewer satisfied with the present study for dual properties.
In addition
Lines 122-123: “Our previous published papers indicate the catalytic study;”. Please, give references. What do you mean by this argument?
Author’s response: The sentence was revised, and a Ref was added.
Lines 217-218: “the absorbance spectrum of CR was measured which exhibiting two peaks at 490 and 970 nm.” Please, support these observations, especially the second peak (970 nm), through adequate literature (in addition to ref. 43) and explanation.
Author’s response: The calibration curve based on the CR concentration and absorbance spectra was added to the text.
Have the authors a rational explanation of the minima/maxima shown in figures 6, 8 and 9?
Author’s response: more explanations have been added to the section 3.4.2.
The format of newly-added references should be checked.
Author’s response: The format was checked.
To conclude, a major revision is still needed prior to final decision. Special attention should be given by the authors to remark i), since a convincing response/additional study is crucial for the full justification of the studies presented, here.
Author’s response: Hopefully that answers to the reviewer's previous questions have clarified the main points of the manuscript and changed the reviewer's perception that our work is in fact a series of ongoing efforts that began with research on photocatalysis and absorption and progressed to include an image-adsorbent cycle process.
In the revised version, all changes and edits to the original manuscript are highlighted in red. We hope that all answers and replies satisfy the reviewer's comments.
We look forward to your kind response.
Sincerely Yours,
Prof. Mohamed Rashad
Corresponding author

Round 3
Reviewer 3 Report
The authors have satisfactorily/clearly responded to all points raised and made adequate changes.
The manuscript can be accepted for publication.
Author Response
Dear Respected Editor of Materials-Journal
Materials-2292521
Thank you for considering our work for publication in your journal, and thanks a lot for the journal staff and you for your efforts. Actually, we answered all the questions raised by the reviewers in the first round. For reviewer 2, we replied to his comments point-by-point, as mentioned below. The corresponding amendments and corrections in the text have been included in the revised manuscript (red marked). We hope that our responses satisfactorily address all the issues raised by the Reviewers.
Sincerely,
Reviewer 3
Dear Respected Reviewer
Comments and Suggestions for Authors
The authors have satisfactorily/clearly responded to all points raised and made adequate changes. The manuscript can be accepted for publication.
Author’s response: Thanks to the reviewer for his positive comment.
